# Investigation of Soft Matter Nanomechanics by Atomic Force Microscopy and Optical Tweezers: A Comprehensive Review

**DOI:** 10.3390/nano13060963

**Published:** 2023-03-07

**Authors:** Alessandro Magazzù, Carlos Marcuello

**Affiliations:** 1CNR-IPCF, Istituto per i Processi Chimico-Fisici, 98158 Mesina, Italy; 2NLHT-Lab, Department of Physics, University of Calabria, 87036 Rende, Italy; 3Instituto de Nanociencia y Materiales de Aragón (INMA), CSIC-Universidad de Zaragoza, 50009 Zaragoza, Spain; 4Laboratorio de Microscopias Avanzadas (LMA), Universidad de Zaragoza, 50018 Zaragoza, Spain

**Keywords:** atomic force microscopy, biopolymers, cellular membrane rigidity, nanoindentation, nanomechanics, optical tweezers, soft matter, stiffness, Young’s modulus

## Abstract

Soft matter exhibits a multitude of intrinsic physico-chemical attributes. Their mechanical properties are crucial characteristics to define their performance. In this context, the rigidity of these systems under exerted load forces is covered by the field of biomechanics. Moreover, cellular transduction processes which are involved in health and disease conditions are significantly affected by exogenous biomechanical actions. In this framework, atomic force microscopy (AFM) and optical tweezers (OT) can play an important role to determine the biomechanical parameters of the investigated systems at the single-molecule level. This review aims to fully comprehend the interplay between mechanical forces and soft matter systems. In particular, we outline the capabilities of AFM and OT compared to other classical bulk techniques to determine nanomechanical parameters such as Young’s modulus. We also provide some recent examples of nanomechanical measurements performed using AFM and OT in hydrogels, biopolymers and cellular systems, among others. We expect the present manuscript will aid potential readers and stakeholders to fully understand the potential applications of AFM and OT to soft matter systems.

## 1. Introduction

Soft matter comprises a large variety of physical systems which can be deformed or structurally altered by mechanical or thermal stresses [1], the aspect that determines soft matter properties [2]. Typical examples of soft matter systems are biopolymers, hydrogels, dendrimers, blends, foams, liquid crystals and biological (cells and bacteria), among others. 

More specifically, we can consider the following systems:Biopolymers are macromolecules composed by the repetition of subunits coming from biological sources. Generally, biopolymers present lower chemical resistance processability and mechanical properties similar to those of synthetic polymers. These peculiarities, combined with their biodegradability efficiency, make biopolymers suitable to be employed in cosmetics [3], in food packaging [4] and in the production of novel medicine compounds [5]. Moreover, the use of biopolymers minimizes the employment of fossil fuels, which is crucial to prevent the undesirable release of greenhouse gases (GHG) during their manipulation and processing. The most promising biopolymers are based on plant cell wall constituents such as the plant cell walls, composed of lignin, cellulose and hemicelluloses, which are highly entangled.Hydrogels are three-dimensional network structures formed by flexible chains interconnected in set ways and swollen by liquid media. Hydrogels can undergo large and reversible expansion or shrinkage under specific conditions which confer their properties. Hydrogels are also employed for the realization of molecular sieves [6], glucose sensors [7], drug delivery systems [8], contact lenses [9], battery binders [10], disposable diapers [11] or bioinks for 3-D printing [12], among others.Dendrimers are three-dimensional branched polymeric macromolecules formed by arborescent construction. Conversely to polymers, where the molecular bond formation is probabilistic, the dendrimer molecule distribution is precise and the chemical bonds between atoms can be accurately described. Dendrimers treasure properties such as self-assembly, chemical stability, low cytotoxicity, polyvalency and good solubility. These properties are relevant in the developing of different fields such as molecular electronics [13], nanomedicine [14], light energy harvesting [15] and catalysis [16].Blends are constituted by homogeneous mixture of two or more polymers called homopolymers and copolymers, respectively, that have been mixed together to produce a new material with different physico-chemical properties. Blends have gained interest due to their ability to modify their mechanical properties. For this reason, blends are exploited for rubber toughening [17], food packaging [18], creation of supports for protein immobilization [19] or design of selective ion-exchange systems [20].Foams are formed by a gas–polymer mixture that provides microcellular structure with inner hollow pores. Foams can be rigid or flexible depending on the geometry of their inner structures. Thanks to their low density, high thermal and acoustic insulation and damping properties, foams are extensive applied for building construction [21], antipollution treatments [22], electronic shielding [23], fuel cells [24] and tissue engineering [25], among others.Liquid crystals are substances flowing like liquids and containing some degree of molecular arrangement ordering. Liquid crystals show excellent electro-optical, reflectance, anisotropic polarizability and low energy consumption qualities. Liquid crystals are widely used as detergent [26], for the realization of displays [27], thermal detection [28] and clinical diagnosis sensors [29].Finally, cells and bacteria are the fundamental anatomical unit of all living organisms and prokaryotic microorganisms that do not bear defined nuclei and generally internal membranous organelles, respectively. Cells can be presented as a biological source for regenerative medicine [30]. Bacteria are suitable as a prototype to fabricate microrobots due to their high motility and convenient controllability [31]. Moreover, the next generation of engineering bioreactors will be focused on cells and bacteria [32].

The impact of exogenous mechanical actions on soft matter systems can have multiple implications, such as the favoring of self-assembly on materials that drive the growth of 2D-crystals [33]; the reduction of wettability on biopolymers, composites and blends by the shrinkage of the inner pore dimensions and morphology [34]; the stiffness increase of plant polymers [35]; or the improvement of the thermal stability [36], among others. This last observation is not unexpected since it has been reported that there is a temperature effect on the decrease of mechanical and tribological properties in polymers [37]. The attention to soft matter materials has grown interest due to their above-described potential performance and their applications. Soft matter materials are actually used and applied to different fields, such as robotics [38], flexible displays [39], tissue engineering [40], design of biosensing devices [41], drug delivery [42], packaging [43] or the optimization of antimicrobial surfaces [44], among many others. The main advantage of soft matter materials is the tunable response under external stimuli such as temperature [45], pH [46], light [47], ionic strength [48] or dynamical flows by microfluidic devices [49], among others. Moreover, working with soft matter systems is appealing since most of them increase the biocompatibility of the evolved material, which is fundamental for biomedical purposes [50]. In this context, many bulk techniques have been devoted to determining the mechanical properties of soft matter, such as multifrequency magnetic resonance elastography (MRE) [51], ultrasonic testing [52], tensile testing [53] and indentation at the microscale level [54]. The main drawbacks of the aforementioned methods are the lack of information of transient phenomena or singularities existing in the tested soft matter samples. Single-molecule techniques such as atomic force microscopy (AFM) and optical tweezers (OT) have been developed to overcome these limitations. 

From its discovery in 1986 [55], AFM has addressed a multitude of physico-chemical properties of soft matter samples. AFM consists of a flexible cantilever ended by ultra-sharp tips that bends when senses external forces [56]. AFM hoards many operational modes to elicit the physico-chemical properties of the sample of interest. AFM imaging assesses the morphological changes that take place upon biomolecular ligand binding and catalysis [57,58,59]. AFM force spectroscopy (AFM-FS) deciphers the adhesion properties [60,61] and the dissociation energy landscapes can be obtained using stochastic dynamic simulations [62]. When low applied forces are acting, the unspecific tip–sample interactions become negligible, and AFM-FS converts to molecular recognition imaging (MRI) [63]. AFM-nanoscale infrared spectroscopy (AFM-nanoIR) covers the chemistry of the bonds involved in the scanned area [64]. Magnetic force microscopy (MFM) analyzes the magnetic response of soft matter samples [65], which could have implications in robots for manipulation and drug delivery [66] or hyperthermia treatments [67]. Scanning electrochemical microscopy (AFM-SECM) is used for electrical characterization [68], which is a crucial step to fabricate more efficient nano-transducers and lab-on-chip biosensor devices [69]. Finally, AFM nanoindentation measure the elastic deformation of the external sample surface when external load forces are exerted. AFM shows many advantages in comparison to other techniques such as the following: (I) The possibility of investigating samples in liquid environments, mimicking the inner cellular conditions allowing in vivo measurements. In this doing, we can use AFM to measure the mechanical properties of the soft matter as function of the pH or ionic strength of the liquid media [70]. (II) AFM measurements can be performed under suitable conditions to preserve the integrity of the investigated sample conversely to cryo-transmission electron microscopy (cryo-TEM), where ultra-low temperatures are required [71]. AFM does not require staining the sample with contrast agents, in contrast to other techniques such as scanning electron microscopy (SEM) or TEM, thus avoiding any artifacts and interferences with the investigated properties [72]. 

OT are scientific instruments that use a highly focused laser beam to trap objects. Coated plastic beads with soft matter materials can be subjected to the underlying attractive or repulsive forces at the piconewton range [73]. OT can quantitatively evaluate the assembly of protein droplets and their characteristics that lead to the pathological solidification [74], biomolecular folding and unfolding events that undergo conformational changes to reach their biological functions [75], decode the molecular mechanisms related to DNA and RNA organization, translation, repair and replication processes [76], hydrodynamic forces involved in endocytosis processes of eukaryotic cells [77], cell migration probed in engineered environments [78] and mechanical response of soft matter systems [79]. Recently, the integration of OT, label-free microscopy, fluorescence spectroscopy and advanced microfluidic systems has allowed the visualization of dynamic interactions in real time [80]. The continuous necessity to launch accurate measurements makes necessary the design of high-throughput modeling toolboxes of rotational torques in OT [81]. Other OT capability overhauls are based on strengthening optical traps with structured illumination, which makes them more sensitive to displacements and increases the resolution of microbead motions on inertial timescales [82]. 

The present review aims to provide the required insights and background to better understand the fundamental basics of AFM and OT and their key role to address the nanomechanical properties of soft matter systems. This work is divided in the following sections: (I) Introduction, (II) Mechanical properties, (III) Non-nanotechnology techniques to determine mechanical properties, (IV) Mechanic models to ascertain Young’s modulus, (V) Working principle of nanotechnology tools to elicit mechanical properties, (VI) Recent examples of nanomechanical properties addressed on soft matter systems and (VII) Discussion and futures perspectives. We expect potential readers and interested stakeholders gain the appropriate knowledge of when to implement AFM and/or OT technologies for the study of soft matter systems mechanics nature at the single-molecule level.

## 2. Mechanical Properties

This section attempts to explain the mechanical properties suitable for measurement in soft matter systems. These include Young’s modulus (*E*), hardness (*H*), viscoelasticity, fracture toughness and energy dissipation or viscosity, among others. Young’s modulus (*E*), also known as elastic modulus, is obtained by the ratio of stress (*σ*) to strain (*ε*) when the shape is recovered by the material after deformation due to an external load force. Stress and strain are proportional in the elastic region following Hooke’s law [83] (Equation (1)):(1)σ=Eε

Thus, Young’s modulus is an intrinsic material property which measures the bond strength between the atoms that form the material of interest. A greater *E* corresponds to stiffer materials with smaller strains. 

Hardness (*H*) is the resistance of a given material to scratching. There are many relative scale methods to classify materials depending on their hardness. The Mohs [84], Vickers [85] and Knoop [86] hardness scales are some of the most implemented today, among others. All of them assess the relative hardness of the material by comparing with standard samples. In mechanics, hardness is devised as the material resistance to permanent deformation under external load forces. The hardness can be estimated in nanoindentation studies (Equation (2)):(2)$$H=\frac{P}{A}$$
where *P* is the exerted load force and *A* is the indentation local area. Unlike Young’s modulus, hardness exhibits a local dependence, with the measured values being different between the material surface and in bulk.

Viscoelasticity refers to the tendency of certain materials to act like both a solid and a fluid. This mechanical parameter is divided by linear and non-linear viscoelasticity response. Linear viscoelasticity is observed when the creep response and the load are separable in the function. The main linear viscoelasticity model was defined by Volterra equations [87] (Equation (3) and Equation (4), respectively):(3)$$\varepsilon \left(t\right)=\frac{\sigma \left(t\right)}{E_{i.creep}}+\int_{0}^{t}K\left(t-t^{\prime }\right)\sigma \left(t^{\prime }\right)dt^{\prime }$$
(4)$$\varepsilon \left(t\right)=\frac{\sigma \left(t\right)}{E_{i.relax}}+\int_{0}^{t}F\left(t-t^{\prime }\right)\sigma \left(t^{\prime }\right)dt^{\prime }$$
where *E_i.creep_* is the elastic modulus for creep, *E_i.relax_* is the elastic modulus related for the relaxation event and *K*(*t*) and *F*(*t*) are the time-dependent creep and relaxation functions, respectively. *E_i.creep_* and *E_i.relax_* are related through their respective time-dependent functions (*K*(*t*) and *F*(*t*), respectively):(5)$$E_{i.creep}=E_{i.relax}\int_{0}^{t}\frac{F\left(t-t^{\prime }\right)}{K\left(t-t^{\prime }\right)}dt^{\prime }$$

Finally, the compliance functions of *E_i.creep_* and *E_i.relax_*, *D*(*t*) and *R*(*t*), respectively, are correlated by the following expression:(6)$$D\left(t\right)R\left(0\right)+\int_{0}^{t}D\left(t-t^{\prime }\right)\frac{\partial R}{\partial t^{\prime }}dt^{\prime }=D\left(0\right)R\left(t\right)+\int_{0}^{t}R\left(t-t^{\prime }\right)\frac{\partial D}{\partial t^{\prime }}dt^{\prime }$$

Linear viscoelasticity is a reasonable approximation for many polymers and ceramics at relatively low temperatures in combination with low stresses.

Non-linear viscoelasticity takes place when the material changes its properties under an exerted load force. The non-linear viscoelastic strain is formulated using the well-known Schapery’s viscoelastic constitutive equation [88,89]:(7)$$\varepsilon \left(t\right)=g_{0}\left(\sigma \right)D_{0}\sigma \left(t\right)+g_{1}\left(\sigma \right)\int_{0}^{t}\Delta D\left(\psi -\psi^{\prime }\right)\frac{d\left(g_{2}\left(\sigma \right)\sigma \left(t\right)\right)}{d\tau }d\tau$$
where the parameters $g_{0}$, $g_{1}$ and $g_{2}$ are function of the strain, $D_{0}=D\left(0\right)$ is the initial value of the creep compliance, $\Delta D=D\left(t\right)-D_{0}$ is the transient component of the creep compliance, *φ* is the strain function associated to nonlinear viscoelastic materials and *τ* is the retardation time. Viscoelasticity properties can be obtained through dynamic mechanical analysis (DMA) [89]. This technique consists of applying a sinusoidal stress and measuring the strain of the material to obtain the complex modulus, the storage modulus and the loss modulus. Often, DMA analyses are carried out for different sample temperature values or different stress frequencies. 

Fracture toughness is the ability of materials to resist the propagation of flaws under an applied stress, with it being assumed that the longer the flaw, the lower the required stress to produce fracture. The stress intensity (*K*) is obtained using the following equation [90]: (8)$$K=\sigma \sqrt{\pi a}$$
where *a* is the crack length. Thus, the fracture toughness is directly proportional to the energy consumed in the plastic deformation (*E*), and the fracture occurs when the stress intensity factor reaches a critical value defined as *K_C_*:(9)$$K_{C}=\sqrt{EG_{C}}$$
where *G_C_* is the strain energy release rate at the critical fracture value point.

Viscous energy dissipation is an irreversible process where the work done by the adjacent layers of the studied material is converted into heat due to shear forces. The dissipation (*ϕ*) is expressed as [91]: (10)$$\phi =2µ\|\frac{\nabla u^{\prime }+{\left(\nabla u^{\prime }\right)}^{T}}{2}\|$$
where *u*′ is the turbulent velocity component, µ is the viscosity of the material and *T* corresponds to the temperature of the adjacent layers. The viscous energy dissipation can be estimated by sophisticated numerical algorithms [92]. Finally, Figure 1 summarizes the four classes of soft matter behaviors under external load forces being divided into elastic (Figure 1a), viscoelastic (Figure 1b), viscoplastic (Figure 1c) and viscous (Figure 1d), respectively. The responses to strain (*ε*) and ε-load time (*t*) reported in Figure 1 are characteristic for each behavior mentioned above and can be used to predict the category of unknown materials by their response to strains. In particular, elastic and viscous materials display a linear slope in stress–strain representations that refer to Young’s modulus and viscosity parameters, respectively. On the other hand, viscoelastic and viscoplastic materials show stress–strain profiles in the form of hysteresis. Viscoelastic events occur when, after the stress is kept out, the strain drops to zero in a time-dependent fashion. Plasticity from viscoplastic phenomena takes place when the strain never comes back to zero after the external load is removed. Thus, the main difference observed between viscoelastic and viscoplastic behaviors is that in the first case the deformation underwent by the material is transitory, whereas in the second one the yield stress is permanent. Both stresses and strains are estimated by finite element method (FEM) software tools which consist of a converged solution for the nodal displacements of post-processing quantities. The study of mechanical properties of soft matter systems is fundamental to better understand their nature, which could assist in finding future potential social and industrial applications. The next sections will provide the fundamental knowledge to the reader about the current available techniques to determine the mechanical properties at bulk and nanoscale levels. Then, the review will focus on the Young’s modulus assessment using nanotechnology tools such as AFM and OT. For this purpose, the existing physical models to extract the elastic modulus of the tested soft matter samples will be discussed depending on the AFM tip geometry and the forces involved during the conducted force–distance curves. 

## 3. Non-Nanotechnology Techniques to Determine Mechanical Properties

### 3.1. Multifrequency Magnetic Resonance Elastography (MRE)

Multifrequency MRE is a non-invasive method where, through algorithmic reconstructions, the stiffness from wave-motion images is resolved under in vivo conditions. Multifrequency MRE investigates the elasticity properties of soft matter samples by applying several frequencies that generate maps of shear modulus (*G*) and the loss angle (*φ*), being convenient for those measurements carried out at a higher range of frequencies. Thus, Young’s modulus (*E*) is calculated by the following expression:(11)E=G (1+υ)
where *υ* is Poisson’s ratio and *G* can be obtained as:(12)$$G=\alpha \;ln\left(\frac{S_{0}}{S_{F}}\right)+\beta$$
where *α* and *β* are calibration coefficients, whereas *S*_0_ and *S_F_* are the baseline and final b-value factors, respectively. The b-values reflect the timing and strength of the gradients employed to create diffusion-weighted multifrequency MRE images. The loss modulus (*G*″) can be calculated through the *φ* and the previously estimated *G*:(13)$$\varphi =tan^{-1}\left(\frac{G^{\prime \prime }}{G}\right)$$

The viscous response of soft matter samples is determined by the extension of *G*″. Multifrequency MRE has been successfully devoted to address the elastic properties of agarose biopolymers [93], hydrogels [94], decellularized pancreatic tissues [95], brain tissues [96] and inflammatory bowel diseases [97], among others. Recently, multifrequency MRE setup was coupled with high-speed cameras to obtain ultrafast images of cellular elasticity [98]. 

### 3.2. Ultrasonic Pulse Testing

The ultrasonic pulse testing technique consists of measuring the velocity of ultrasonic pulses passing through soft matter materials. Ultrasonic pulse testing setup includes an electronic circuit to generate tunable pulses and a transducer to transform electronic signals to mechanical pulses by defining the oscillation frequency as close-feedback and a pulse reception circuit that receives the signal for the further processing. Higher pulse velocities take place when the measured elastic properties are large. Otherwise, low pulse velocities indicate poor mechanical performance of the studied material of interest. The dynamic Young’s modulus is deciphered by the following equation [99]:(14)$$v=\sqrt{\frac{E\left(1-µ\right)}{\rho \left(1+µ\right)\left(1-2µ\right)}}$$
where *v* is the ultrasonic pulse velocity, µ is the dynamic Poisson’s ratio, *ρ* is the density of the soft matter measured material. The only weak point that ultrasonic pulse testing technology depicts is the requirement of regular surfaces to enable the accurate measurement of the pulse velocities. Dynamic elastic modulus has been assessed by ultrasonic pulse testing for thermoplastic polymers [100], photo-clickable poly(ethylene glycol) hydrogels [101], composite blends [102], syntactic foams reinforced with hybrid fibers [103], liquid crystals [104], biological tissues [105] such as cortical bone [106] or pulmonary capillary tissues [107]. 

### 3.3. Tensile Testing

Tensile testing is a destructive process that delivers information about the tensile and yield strengths and ductility of soft matter materials by measuring the force requested to stretch the material of interest until it reaches its rupture point. There are several approaches to establish the aforementioned material mechanical properties. The Hollomon equation correlates the true-strain–true-stress curves [108]. The Hollomon’s equation shows some constraints [109], such as the impossibility to characterize the tested sample in the full strain–stress range due to the observance of distinct hardening stages. For this reason, alternative models have been hypothesized such as the Swift [110] and Voce [111] equations. The Swift equation is more appropriate to describe the stress–strain profiles in small strain regions, whereas the Voce expression is the most suitable to predict the stress–strain curves in large strain regions. Table 1 outlines the above-described calculation methods for tensile testing measurements.

### 3.4. Indentation (Macroscale Level)

Macroindentation tests were introduced to determine the hardness of materials. Indenters with several geometry shapes and fabrication compounds have been tested to optimize the experimental results, the most commonly employed being the spherical diamond [127], square-based diamond pyramid [85] or rhombohedral-shaped diamond indenter [86]. Square-based diamond pyramid indenters are the selected to conduct the Vickers hardness test [128]. This method is based on the high resistance of the indenter to self-deformation. The Vickers pyramid number (*VPN*) is deciphered as:(15)$$VPN=\frac{F}{A}$$
where *F* is the load force and *A* corresponds to the surface area resulting indentation. *A* can be elicited by the following expression:(16)$$A=\frac{d^{2}}{2\;\sin\left(136{}^{\circ}/2\right)}$$
where *d* is the average length of the diagonal left by the indenter. Vickers hardness test presents some limitations, such as the experimental acquisition speed, and is not completely accurate for small-size objects due to the large indenter impression. To overcome the aforementioned drawbacks, sharp Berkovich triangular pyramid or sphere-shaped indenters were designed to extrapolate the hardness and elastic modulus from the curves of indentation load corresponding to displacement coordinates [129]. Strain hardening and Young’s modulus of the tested soft matter samples are obtained from the maximum load and the initial unloading slopes. The main limitation of the above-described non-nanotechnology tools is their inability to observe singular events or mechanical gradients in bulk measurements. Furthermore, the continuous necessity of measuring the elastic properties of specific local areas drove the progress of single-molecule techniques in the contribution to the physical models described in the next section. Indentation measurements have been used to study the mechanical properties of polyethylene [130] and chitosan [131] polymers, hydrogel coatings [132], acrylonitrile butadiene styrene and polycarbonate blends [133] or smart composites [134], liquid crystals [135] and bone tissues [136].

## 4. Mechanic Models to Ascertain Young’s Modulus

This section explains all the existing theoretical frameworks to extract the elastic modulus of soft matter samples by nanotechnology tools, overall by AFM, where the tip apex works as a nanoindenter. Here, the most optimal conditions required to use each model in combination with their potential limitations are addressed.

### 4.1. Hertz Model

The Hertz model does not account for adhesion forces established between the nanoindenter and the external sample surfaces [137]. The Hertz model assumes the nanoindentor is a perfect sphere that causes a perpendicular penetration to a perfectly planar surface. The second presumption is that the strain–stress relation is linear, following the Hooke’s law (Equation (1)). Then, Young’s modulus can be calculated through the load force (*F*):(17)$$E=\frac{3FR\;}{4a^{3}}$$
where *R* is the sphere radius of the nanoindenter and a is the contact radius between both surfaces. The main shortcomings found in the Hertz model are the following: (I) Nanoindentation probes are not perfect spheres. Furthermore, the indentation is not made perfectly perpendicular to the surface since the probes are slightly tilted by 10–15 degrees. (II) Most of the soft matter systems show some viscoelastic response when they are indented. This is the main reason of the hysteresis observed in the strain–stress profiles (Figure 1b). To minimize this unwanted viscoelasticity behavior displayed by the measured sample, we can decrease the indentation rate since viscosity increases with the indentation speed, as reported in living cell systems [138]. 

### 4.2. Johnson, Kendall and Roberts (JKR) Model

The JKR model takes into consideration the short-range forces located in the surface contact area between the nanoindenter and the external sample surface [139]. This model is the most suitable for large spherical indenters that expose greater contact areas, and thus causing strong adhesion events. The JKR model accounts the transfer of work from the contact zone to the interaction sphere being equal to wd(π*a*^2^). The elastic modulus can be estimated by the following expression: (18)$$F=\frac{4\;E\;a^{3}\;}{3\;R}-2\sqrt{2\pi Ewa^{3}}$$
with *w* being the required energy to separate a unitary area of both surfaces. The main limitation encountered in the JKR formulation is the strong dependence on surface slopes during the load forecasting. For this reason, there are no adhesion events in the fractal limits. The JKR model also provides inaccurate data when boundaries of non-fully detached surface contact are evaluated. For all the above-described reasons, the JKR model is suitable to gather the mechanical cues of soft matter systems for large elasticity parameters (*λ*). 

### 4.3. Derjaguin, Müller and Toporov (DMT) Model

The DMT model is applied when long-range surface interactions outside the contact area, such as Van der Waals forces, are considered [140]. The Van der Waals interactions located at the perimeter of the indented surfaces leads to additional attraction faced between the external sample surface and the probe. This approach is valid for small-size spherical indenters, stiff materials and relatively weak interactions between both of them. It is supposed that the geometry of the deformed surfaces is closely measured to provide a solution to the Hertz model’s limitations. The Young’s modulus of studied soft matter materials is obtained by:(19)$$E=\frac{R\;F\;\;}{a^{3}}\left(F+2\pi Rw\right)$$
where the required parameters are described above. The capital restrictions experienced in the DMT model are the possibility to reduce the contact area based on the limited indenter geometry. For this reason, DMT formulation only can be applied to small *λ*.

### 4.4. Maugis Model

As aforementioned, the Hertz, JKR and DMT models show some limitations. The Hertz model neglects the adhesion contribution between both contacting surfaces, JKR formulation only accounts the inside contact area, whereas the DMT model takes into consideration the outside surface area. For this reason, the JKR and DMT models are valid for indenters with large radius and a combination of short radius/soft matter samples, respectively. In order to overcome the above-described weaknesses, the elastic deformations suffered by the sample of interest were assessed as a function of *λ* parameter [141]:(20)$$\lambda =\frac{2.1\;}{D_{0}}\sqrt[{3}]{\frac{Rw^{2}}{\pi E^{2}}}$$
where *D*_0_ is the interatomic distance. The Maugis theory is located on the verge between the DMT and JKR models and is the least employed due to the complexity of its equations and the impossibility to obtain Young’s modulus by experimental force curves. In particular, the complexity is mainly due to the self-recursive relation of the elastic modulus that depends on *λ* and vice versa. Figure 2 depicts the most suitable regions regarding the relation established between load force and *λ* parameters to use all model frameworks that aim to accurately obtain the elastic modulus of soft matter systems. 

It is noteworthy that rigid bodies comprise the region where the values of load forces and λ are low. The Bradley model studies the elasticity performance of rigid entities [142] which are not discussed here because this review work is focused on soft matter systems. We can conclude that the best model to fit experimental nanomechanical data from soft matter systems is the DMT model. Nevertheless, the Hertz, JKR and Maugis models could also be considered depending on the experimental conditions.

## 5. Working Principle of Nanotechnology Tools to Elicit Mechanical Properties

### 5.1. Atomic Force Microscopy (AFM)

Atomic force microscopy has proven to be an excellent approach to address the mechanical properties of soft matter systems at the nanoscale level, where the AFM tip works as a nanoindenter. A proper AFM tip characterization is required before running a nanomechanical experiment in order to prevent the acquisition of non-trustful raw data. In Figure 3, all the mandatory steps to be conducted in this regard are shown. First, the tip radius needs to be accurately characterized. The measurement of standard samples facing homogeneous conical-shape features by AFM imaging allows the precise determination the AFM tip radius after convolution analysis (Figure 3a). Individual local cone peaks in the scanned topography image are successively examined by measuring the slope away from the peak in all directions, and thus assessing the AFM tip sharpness. 

The effective tip diameter (ETD) is defined as the diameter of a circle containing the same area with respect to the measured tip cross-section is achieved using this strategy. All the aforementioned aspects significantly aid the potential users in the decision making regarding whether the AFM tip is acceptable for use. Then, the deflection sensitivity of the AFM cantilever can be obtained by calculating the curve-slope average of at least three force–distance curves on stiff solid surfaces (Figure 3b).

Nevertheless, the determination of accurate deflection sensitivity values is not as straightforward as it seems, rendering an estimated error of around 30%. This fact, based on the calculation of deflection sensitivity, is related to the cantilever spring constant, which in turn is established by the free-resonance cantilever thermal spectrum. To overcome this situation, a recent approach named standardized nanomechanical atomic force microscopy procedure (SNAP) [143] was developed. This approach consists of calculating a correction factor (ζ) for the deflection signal when the cantilever spring constant is more precisely calculated by using a vibrometer. SNAP was successfully employed to assess the elastic modulus of polyacrylamide gels, minimizing the aforementioned standard error from 30% down to 1%. If the spring constant of the AFM cantilever is expected to be more than 1 N/m, the most convenient solid surface to conduct the force–distance curves is sapphire, whereas freshly cleaved mica surfaces can be employed for those AFM cantilevers with spring constants lower than 1 N/m. 

Finally, the thermal noise accounts for the AFM cantilever stiffness in the direction of piezoscanner movement—that is, perpendicular to the oscillating movement—when the AFM setup does not provide excitation to the AFM probe (Figure 3c). The thermal noise method is rooted in the equi-partition theorem (Equation (21)), which combines the spring constant of the cantilever with the associated Brownian motion [144]. This fact is due to the assumption that the AFM cantilever acts as an ideal harmonic oscillator.
(21)$$\langle \frac{1}{2}\;k\;Z^{2}\rangle =\frac{1}{2}k_{B}T$$
where *k* is the spring constant of the AFM cantilever, *Z*^2^ is the mean square motion of the AFM cantilever, *k_B_* is the Boltzmann constant and *T* is the kelvin temperature. Thus, spring constant values are calculated through the mean square AFM cantilever displacement that can be pinpointed by integrating the power spectral density (PSD, Figure 3d). There exist many applicable theoretic models available in fitting the PSD response. Commonly, simple harmonic oscillator (SHO) and Lorentz fitting models are used for PSDs with low and large Q-factors, respectively. The Q factor of an AFM cantilever indicates its capacity to dissipate energy and thus the existing damping. AFM cantilevers with higher Q-factors allow to increase the force sensitivity. The recommendation is to employ SHO and Lorentz fitting models in liquid and air environments, respectively, because the damping of the AFM cantilever by the liquid molecules causes lower Q-factors on the PSD. This observation is based on the fact that the AFM cantilever motion drags the neighboring liquid molecules, thus leading to a strong increase in the cantilever effective mass by one magnitude order factor. This phenomenon triggers stronger hydrodynamic interactions between the cantilever and the liquid molecules, rendering lower Q-factors of ~4–10 times lower compared with measurements conducted in air conditions. The model used can also depend on the cantilever spring constant value range. The model used for soft AFM cantilevers with spring constants *k* < 1 nN/s is the thermal tuning [145]. On the other hand, stiff AFM cantilever spring constants *k* > 1 N/m are calibrated by the Sader’s method [146].

In Table 2 are reported the best options to choose the AFM cantilever spring constants according to the intended elastic modulus of the measured soft material. The selection of the AFM probe is based on the fact that it needs to cause enough deformation of the soft matter sample and still retain a high force sensitivity. 

Therefore, there exists a compromise between image resolution and Young’s modulus acquisition accuracy, so stiff AFM probes may improve Young’s modulus correctness at the expense of sample damaging. For accurate elastic measurements, at least 2 nm of sample deformation is requested without causing permanent damage. 

Before starting experiments to address the nanomechanical properties of soft matter systems, it is requested to follow all the above-described steps to accurately calibrate the AFM tips and thus prevent the acquisition of misleading raw data. AFM is capable of recording force–distance curves (Figure 4) at local sample areas discriminating between different mechanical properties.

During a single force–distance curve cycle, the AFM tip approaches the external sample surface until the contact between both bodies occurs (Figure 4a). At this point, the rigid AFM tip apex causes a deformation of the indented surface. Then, a different direction of movement takes place, and the tip moves away the sample. Adhesion forces between the tip and the sample emerged during the contact being broken at sudden retract point (Figure 4b). The tip will move up to the original position defined at the start of the cycle. The energy dissipation and sample stiffness can be estimated once the force–distance curve is recorded. The area between the approach and retract curves determines the energy dissipation (Figure 4c), whereas the stiffness relies on the slope of the force–distance curve (Figure 4d). The stiffness of the soft matter can be quantitatively converted to Young’s modulus through the mechanic models described in Section 4, which are selected depending on the geometry of the AFM tip and the nature of the forces that dominate during the contact. For all the aforementioned reasons, the accurate characterization of the used AFM tip is crucial to precisely determine the Young’s modulus of soft matter systems. 

Finally, an array of force–distance curves over specific sample surface areas can lead three-dimensional maps where it is possible to simultaneously compare the topography of the scanned area with the elasticity map. This aspect allows to show local variations in surface Young’s modulus. 

### 5.2. Optical Tweezers (OT)

The first idea that light could exert a force on matter dates back to the 17th century, when Johannes Kepler argued that the dust tail of comet is due to the radiation pressure exerted by the sun’s rays on sublimated components of a comet. The radiation pressure is the mechanical pressure exerted upon any surface due to the momentum exchange between photons and matter. The momentum carried and exchanged by photons is so extremely small that radiation pressure and its application became very interesting only after the invention of the laser. In fact, it was the 1987 when Arthur Ashkin (Nobel Prize in Physics in 2018) developed optical tweezers (OT) thanks to his pioneering experiments on the interaction of laser light with microparticles [147,148,149]. Since their invention, OT have become a key tool for the contactless manipulation and characterization of a wide variety of objects, such as atoms [150], nanoscopic [151] and microscopic particles [151], as well as viruses, biomolecules, bacteria and cells [149,152,153,154,155]. Optical tweezers consist of a tightly focused laser beam able to exert optical forces on micro and nano-objects as a consequence of the conservation of the linear momentum in the light–matter interaction [156]. Under proper experimental conditions (i.e., laser power, size of the particle and refractive indexes of both the medium and the particle), optical forces are able to confine a particle near the focal spot of a focused laser beam [156]. Figure 5a depicts the main components existing in OT setups.

A full understanding of optical forces requires the full electromagnetic theory describing the light–matter interaction based on the Maxwell’s equations [157]. However, some simplifications and approximations, depending on the particle size, have been made to provide an easier understanding and physical insight of optical forces [151]. In particular, for particles larger than the wavelength of the trapping beam, we can use the Geometrical Optics (GO) approximation, where the incoming optical field, generated by the focused laser beam, can be considered as a collection of light rays carrying a portion of the total optical power and linear momentum (Figure 5b). When a ray impinges on a particle, it will be partly transmitted and partly reflected, according to the Snell’s law [158]. If for simplicity we consider that the portion of ray reflected is very small and negligible, an incident ray (${\vec{R}}_{\mathrm{inc}}$) impinging on a particle will be completely transmitted through the particle ${\vec{R}}_{\mathrm{tran}}$ and then refracted outside the particle (${\vec{R}}_{\mathrm{out}}$), as shown in Figure 5b. During these events, a certain amount of momentum, $\Delta \vec{P}={\vec{P}}_{\mathrm{inc}}-{\vec{P}}_{\mathrm{out}}$, is exchanged between the ray and the particle, where ${\vec{P}}_{\mathrm{inc}}$ and ${\vec{P}}_{\mathrm{out}}$ are the momenta associated with the incident and the outgoing ray ${\vec{R}}_{\mathrm{inc}}$ and ${\vec{R}}_{\mathrm{out}}$ respectively [159]. The exchanged momentum $\Delta \vec{P}$ during a time interval Δt gives rise to an optical force $\vec{F}=\Delta \vec{P}/\Delta t$ according to the Newton’s second law (Figure 5b). In the GO approximation, the optical force $\vec{F}$ acting on a particle is the sum of all the momenta exchanged by all the rays impinging on the particle. 

Optical forces can be separated into two different components: gradient forces ${\vec{F}}_{\mathrm{grad}}$, proportional to the light intensity gradient in the focal region and scattering forces ${\vec{F}}_{\mathrm{scat}}$ proportional to the light intensity (Figure 5b). Gradient forces pull the particle towards the focal spot, while scattering forces, due to the radiation pressure, push the particle along the direction of the laser beam propagation [151]. Optical trapping is achieved only when gradient forces overcome the destabilizing effect of the scattering forces [156,159,160]. For small displacements x of the trapped particle from its equilibrium position $x_{\mathrm{eq}}$, a restoring optical force $F_{x}$, linearly proportional to the displacement $\left(x-x_{\mathrm{eq}}\right)$, brings the particle back to its equilibrium position, acting like a Hookean spring with a fixed stiffness $k_{x}$ (Equation (22)):(22)$$F_{x}\approx -k_{x}\left(x-x_{\mathrm{eq}}\right)$$
where, for simplicity, we consider displacements only in one dimension (Figure 5c).

It is noteworthy that the optical trapping potential, resulting by the integration of $F_{x}$ over the displacements *x*, is harmonic (Equation (23)):(23)$$U_{x}=\frac{1}{2}\;k_{x}{\left(x-x_{\mathrm{eq}}\right)}^{2}$$

The equilibrium position $x_{\mathrm{eq}}$ corresponds to the minimum of the optical potential, where there is no net force acting on the particle (Figure 5c). 

The optical trap stiffness $k_{x}$ can be calculated by several calibration methods analyzing the trajectory of the trapped particle, which can be obtained using digital video microscopy (DVM) or a quadrant photo diode (QPD) [156,161]. Furthermore, by calibration we can also obtain the calibration factors pixel to µm by DVM for a camera and volt to µm for the electric signal from a QPD. These calibration factors allow us to measure sizes and displacements of the trapped objects with proper units of measurement [156]. Calibrated OT can be also employed as a force transducer for photonic force microscopy (PFM). Once $k_{x}$ is obtained, an external force, $F_{\mathrm{ext},x}$, acting on a trapped particle can be quantified by measuring the displacement $\Delta x_{\mathrm{eq}}=\left(x-x_{\mathrm{eq}}\right)$ of the particle from its equilibrium position $x_{\mathrm{eq}}$, i.e., $F_{\mathrm{ext},x}=k_{x}\Delta x_{\mathrm{eq}}$, as indicated in Figure 5c. Using OT, we can measure an external force acting on a trapped bead and the resulting displacement from its equilibrium position. Moreover, OT can exert optical forces on soft matter systems and are currently employed to study the elastic properties of these systems by measuring the force required for their stretching. See Section 6.2 for more details.

## 6. Recent Examples of Elastic Properties Addressed on Soft Matter Systems

### 6.1. Atomic Force Microscopy (AFM) to Evaluate Elastic Properties of Soft Matter

This section sets forth relevant soft matter systems where their Young’s modulus were assessed using AFM nanotechnology tools. Table 3 summarizes recent examples in this regard. It is remarkable the strong impact of the environmental conditions on Young’s modulus as rooted in the next several examples. The first case is based on the almost 2-fold decrease of Young’s modulus values found in *Staphylococcus epidermis* when the ionic strength increases (from deionized water to 100 mM CaCl_2_) [162]. The same outcome was observed for *Escherechia coli* cells when the molarity of KNO_3_ salt solution increases from 1 mM to 100 mM, leading a decrease of elastic modulus nearby of three times, from 950 kPa to 300 kPa, caused by the exo-osmotic water loss [163]. The fixation procedures employed can also impact on the gathered Young’s module values. One illustrative example is the case of human cancer colon cells where Young’s modulus varies from 0.4 kPa to 309.5 kPa (almost 10 times greater) when the tested cancer cell sections are obtained by frozen non-fixed protocols and embedded by paraffin, respectively [164]. Additionally, users need to pay attention to all the settings involved during data acquisition. Table 3 depicts those parameters such as the load force exerted to the AFM tip or the indentation depth [165] that affect the gathered Young’s modulus values. The case of the load force is especially interesting since the geometry of the AFM tip is the main factor that can alter these exerted forces by several orders of magnitude. As mentioned above, sharp tips display load forces in the range of nN, whereas for AFM levers ended with microbeads, the load forces increase up to the µN scale. On the other hand, higher indentation depths render greater Young’s modulus values. Based on the high sensitivity exhibited by AFM setups, it is mandatory to well define the experimental conditions before launching the data acquisition and to keep them systematically constant during the different tested soft matter samples of interest. Taking this point into consideration, AFM can unravel the Young’s modulus of soft matter systems evaluating the changes on sample nature such as the prognosis of the status of the human skin [166] or the bone cells [167] in patients, environmental conditions in biopolymers and composites, such as the changes of R.H [168] and temperature [43], respectively, or the footprint of pH on hydrogels [167]. For all the above-described aspects, we can conclude that it is extremely important the good practices are followed by users in determining the elastic properties of soft samples. The knowledge provided in this section will be useful not only for beginners but also for advanced users to meet all required control settings during the nanomechanical AFM measurements and also to identify the potential shortcomings which can appear during data acquisition. 

In Figure 6, the range of elastic modulus values found by AFM for soft matter systems is reported. The observation of an enormous range of elastic properties is evidenced. Hydrogels and living cells exhibit the larger distribution of Young’s modulus values (from ~2 kPa to ~900 MPa). This fact is based on the different polymerization degree and cellular maturation of different cell lines, respectively. The elastic modulus of tissues (from ~900 Pa to ~1200 MPa) strongly differs from biopolymers (from ~1 GPa to dozens of GPa), dendrimers (from ~0.1 GPa to ~1 GPa), blends (from ~1 GPa to ~10 GPa), foams (from ~1 GPa to ~10 GPa) and liquid crystals (from ~1 GPa to ~10 GPa). The larger elastic modulus suggests higher intramolecular cohesion between the neighboring atoms which conform the soft matter systems. These greater cohesive forces render more resistance of the tested materials under external load forces, and thus their mechanical performances are improved.

### 6.2. Elasticity of Soft Matter Systems Addressed by Optical Tweezers (OT)

Calibrated OT can be also used to study the elasticity of soft matter system, ranging from synthetic polymers to cells and biological molecules, such as red blood cells and DNA. The elasticity of the DNA plays a crucial role in all its processes e.g., folding, recombination, replication and transcription, etc. Moreover, free molecules of DNA represent an excellent experimental model used to study and validate theoretical models of polymers [181]. A single DNA molecule, for small extension beyond its rest limit, can be approximated like an ideal entropic spring (a polymer chain subjected to thermal fluctuation), with a specific length, $l_{0}$, indicating its maximum physically possible extension, a persistence length, $l_{p}$, quantifying its bending stiffness and a stiffness, $k_{0}$, when stretched slightly beyond $l_{0}$ [182]. These parameters, characterizing the flexibility and elasticity of a DNA molecule, can be obtained by stretching a single molecule with OT and fitting the force–extension curve with a theoretical model. Experimentally, one end of DNA can be attached to a fixed microscopic bead or anchored to a cover glass, while the other end is attached to an optically trapped bead, as shown in Figure 7c [179,180]. Pulling the trapped bead connected with one end of DNA with OT, we can stretch the polymer, measuring the corresponding stretching force.

For small stretching, when the end-to-end extension is shorter than $l_{0}$, the force is small because only the bends of the polymer are removed. Here, the DNA behaves like an entropic spring well described by the worm-like chain (WLC) model, and its elasticity is represented mainly by $l_{p}$ [183,184,185]. When the end-to-end extension approaches $l_{0}$, there are no more bends to be removed, the DNA shows an elastic stiffness $k_{0}$ and the force increases with the stretching of the polymers according to the WLC model, where $k_{0}$ and $l_{p}$ are proportional to each other. Unfortunately, the inextensible WLC model fails when these parameters are measured in presence of cations (i.e., as function of sodium ion Na^+^ concentration). In particular, the force–extension curves measured do not follow the model predictions, and $k_{0}$ and $l_{p}$ are no longer proportional to each other. This discrepancy arises because the inextensible WLC model does not take into account electrostatic repulsion between the partially screened phosphates on the negatively charged backbone on DNA [186]. According to the experimental force–extension curves obtained by stretching a DNA molecule with OT, it was observed that a DNA molecule can be stretched easily ($k_{0}$ decreases) as the salt concentration is lowered, but its bending stiffness, $l_{p}$, increases. This leads to new models for bending and stretching properties of DNA and for its structure during overstretching at not too low salt conditions [182,187]. Thanks to these new models, it was possible to calculate the DNA elasticity parameters for different buffer solutions (Table 4), increasing Na^+^ ions concentrations (Table 5) and increasing the ionic strength of increasing NaCl concentrations (Table 6). The values reported in the tables were obtained by fitting the experimental force–extension curves with different theoretical models [181,182,186]. 

The Young’s modulus of λ-bacteriophage DNA was calculated for the high salt case reported in Table 6, with k_D_ = 1400 pN, yielding a Young’s modulus value of E = 450 MPa [186]. The elasticity characteristic of each living cell plays a dominant role in its biological functions. Often, variation of the characteristic elasticity of the cells can lead to human diseases, whose progresses and identification, in some cases, can be performed with OT [188,189]. Furthermore, studying the mechanical response of the cells using OT can be used to develop quantitative models for their mechanisms of deformation within the human body. A human red blood cell (RBC) has a biconcave shape and an averaged diameter of 8 µm, its lifetime is about 120 days, during which time it circulates through the human body almost half a million of times. The oxygenation of the body also depends on the elasticity of RBCs, which deform during the microcirculation of the blood to be able to go through capillaries as small as approximately 3 µm. The elasticity of RBCs and their adhesiveness with other cells can be altered by the malaria intra-cellular parasite, losing its ability to undergo large deformations [188,190]. Every year, 2–3 million of people die from malaria. Furthermore, the deformability of RBCs can also be altered by different types of diabetes such as diabetic kidney disease and type 2 diabetes mellitus (T2DM) [189]. The elasticity and deformation of an RBC can be measured by OT, either with its extremities attached to two beads, similarly to the method shown for DNA in Figure 7a,b, or directly trapped without any beads, as in Figure 7c [188,189,191]. Direct trapping and stretching of the RBC is easier since there is no need of a microfluidic chamber and to attach the RBC to beads, and this tapping is still able to exert force in the order of pN [189,191]. The two extremities of an RBC can be directly trapped and stretched by a dual beam OT, keeping the position of a trap fixed and moving the other one after verifying that the optical power delivered is not altering the properties of the trapped RBC [189]. The deformability of an RBC can be quantified by the deformability index (DI), defined as:(24)$$\mathrm{DI}=\frac{\mathrm{final}\;\mathrm{streched}\;\mathrm{length}\;\mathrm{of}\;\mathrm{RBC}-\mathrm{initial}\;\mathrm{streched}\;\mathrm{length}\;\mathrm{of}\;\mathrm{RBC}}{\mathrm{initial}\;\mathrm{streched}\;\mathrm{length}\;\mathrm{of}\;\mathrm{RBC}}$$

The investigation of RBC deformability in T2DM with and without diabetic retinopathy (DR) indicated that size and reduced deformability of RBCs play a crucial role in microvascular complications (Table 7) [189].

The mechanical properties of the RBC were also studied with an optical stretcher, which is an OT-based setup, able to trap and stretch objects using optical forces. In particular, the value of the product of Young’s modulus, *E*, and the membrane thickness, *h*, was found as *Eh* = 3.9 ± 1.4 · 10^−5^ Nm^−1^ [192]. 

Optical tweezers can also transfer angular momentum from a polarized laser beam to birefringent or chiral trapped particles inducing toques on them and causing them to rotate [156,193,194,195,196]. Recently, an OT-based setup was used to measure the shear viscosity of the macropinosome of a living macrophage cell in vivo by transferring spin angular momentum to a vaterite birefringent micro-particle internalized by the cell, Figure 7d [194]. This transfer generates an optical torque defined by $\tau_{\mathrm{opt}}=\Delta \sigma P/\omega$, where P is the trapping power, Δσ indicates the change in the degree of circular polarization and ω is the angular frequency of the trapping laser beam. The generated torque will induce continuous rotation of the vaterite particle. The torque exerted by the fluid on a rotating sphere is $\tau_{\mathrm{drag}}=8\pi \eta a^{3}\Omega$, where η is the shear viscosity of the fluid surrounding the sphere, a is the radius of the sphere and Ω is the angular rotation frequency of the sphere. The two drags can be equated, and the shear viscosity can be calculated as [197,198]: (25)$$\eta =\frac{1}{8\pi a^{3}}\left(\frac{\tau_{\mathrm{opt}}}{\Omega }\right)$$

Equation (21) allows the calculation of the shear viscosity of a fluid by a direct measurement of *P*, Δ*σ*, *a* and Ω. In particular, the shear viscosity for a macropinosome lumen inside a living cell was calculated as *η* = 1.01 ± 0.16 mPa s. This value is similar to the measured shear viscosity *η* = 1.05 ± 0.02 mPa s of the L-15 culture medium used during the experiments. This similarity can be explained because a macropinosome mostly consist of internalized surround medium, in this case L-15 culture medium [197]. 

## 7. Discussion and Future Perspectives

Soft matter encompasses a multitude of systems such as biopolymers, hydrogels, dendrimers, blends, foams, liquid crystals and living cells, bacteria or viruses, among others. The comprehension of soft matter systems is crucial since they have high potentiality for practical applications. Versatile soft matter could act as core actor to develop key future technologies including electronics [199], water purification membranes [200], tissue engineering [201], artificial intelligence robots [202] or the design of efficient therapies against human diseases [203]. For this reason, the knowledge of their intrinsic properties can significantly aid to predict their performance under certain conditions of interest. Soft matter is characterized by its structural and dynamic complexity, which confers its mechanical properties. In this context, nanotechnology tools can furnish accurate data to better understand the response of soft matter systems under external load forces. This review is focused on the use of AFM and OT to decipher the elastic properties of soft matter. The working principles of both techniques are detailed to significantly aid beginners and stakeholders to make the best decision during the experiments and the subsequent raw data analysis. This aspect will serve to unequivocally obtain the elastic modulus of soft matter systems. This work also showcases the theoretical model frameworks built for AFM nanomechanical experiments, being fully explained which model is the most convenient to use according to the AFM tip apex geometry. Furthermore, recent examples are presented to the reader in order to be aware of the excellent opportunities offered by both techniques in this field while giving a glimpse into efforts made by researchers dedicated to investigate highly functional soft matter materials.

Promising avenues of research are opened by combining AFM/OT with other operational mode techniques and thus expand the acquaintance of tested soft matter systems [204,205]. AFM nanoindentation has successfully coupled with AFM-nanoIR to study human hair bundles [206], with total internal reflection fluorescence (TIRF) to correlate the mechanical interactions with fluorescence dynamics during the damage of bacterial cell walls [207] or elicit mechanotransduction processes of cells immobilized on electrically stretchable substrates [208]. Cellular mechanics under the effect of the shear forces can also be evaluated, integrating microfluidic devices to AFM [209]. In particular, this technology has successfully devoted to the nanomechanical characterization of circulating tumor cells (CTCs) [210]. Finally, tip-enhanced Raman spectroscopy (TERS) measurements can be simultaneously acquired with nanoindentation experiments [211]. This approach has allowed to measure the bimodal complementary compositional and elasticity of bone implants [212]. 

On the other hand, OT can be also combined with other correlated techniques to unravel many properties of soft matter. The kinetics can be assessed by multicolor epi-illumination fluorescence microscopy [213]. This multimodal microscopy can be successfully used in the study of single cell nanomechanics while imaging their response through fluorescent labeled protein force-transductors [214]. Fluorescence excitation lasers can be also coupled with OT, next to the fiber laser of the optical trap to render confocal fluorescence, and thus measuring the bind of single fluorophores [215]. Furthermore, OT can also be coupled with total internal reflection fluorescence (TIRF) microscopy that allows the measurement of lateral movements of trapped organelle bodies located inside the living cells in real time [216]. The coupling of Raman spectroscopy with OT gives rise to the new technique of Raman tweezers (RT), which allows us to obtain the fingerprints of the single trapped objects without any shielding effects due to other particles or the substrate. This technique has been successfully employed to study the effect of thalassemia on hemoglobin deformability [217]. Moreover, RT was also used to trap and chemically analyze individual tire and road wear particles in liquid environments [218], detect microplastic polymers in seawater [219] and aid in the identification and subsequent classification of marine bacteria based on their cell phenotypes [220]. Recently, RT was also used to investigate single grains of cosmic dust [196]. Finally, force-induced mechanical balance data of soft matter obtained by OT can be compared by complementary techniques such as micropipette aspiration system with stimulated emission depletion (STED) microscopy [221]. Coupling AFM and OT with other setups displays many advantages in comparison with other techniques such as förster resonance energy transfer (FRET)-based biosensors to screen mechanical parameters [222]. The sensitivity and selectivity limitations of this FRET-biosensor platforms restrict their use in the field of nanomechanics, being thus more broadly exploited for sample imaging. 

The combination of experimental AFM and OT measurements with mathematical modeling and computational methods is expected to open promising prospects on collecting more robust data on soft matter nanomechanical properties. In this context, the tested sample is divided in specific regions in order to generate accurate 3D models for subsequent numerical simulations. Recently, multiscale modeling approaches have been devoted to determining the mechanical parameters of bone tissues [223,224] or to predict by machine learning the response of soft matter to specific stresses, paving the way to the discovery of novel classes of complex and novel behavior regimes [225]. These computational simulations can complement nano (AFM or OT) and the macroscale (stated in Section 3) mechanical properties of soft matter systems such as alveolar cells [226], being fully extendable for other potential samples of interest. 

## Figures and Tables

**Figure 1 nanomaterials-13-00963-f001:**
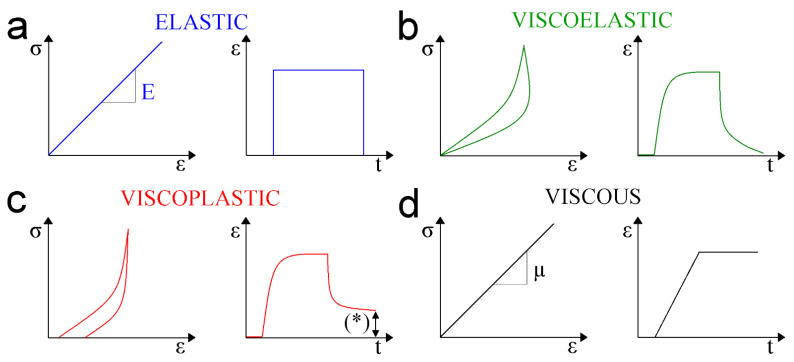
Schematic representation of (**a**) elastic, (**b**) viscoelastic, (**c**) viscoplastic and (**d**) viscous mechanical properties characterization. The left-hand plot depicts stress (σ)–strain (ε) relationships for all types of soft matter biomechanics. The slopes obtained in the elastic and viscous solids represent the Young’s modulus (E) and viscosity (µ) of the material, respectively, whereas the right-hand plot illustrates the respective correspondence between strain (ε) and time (t) for each mechanical performance above described. (*) corresponds to residual strain not recovered in those viscoplastic deformation events.

**Figure 2 nanomaterials-13-00963-f002:**
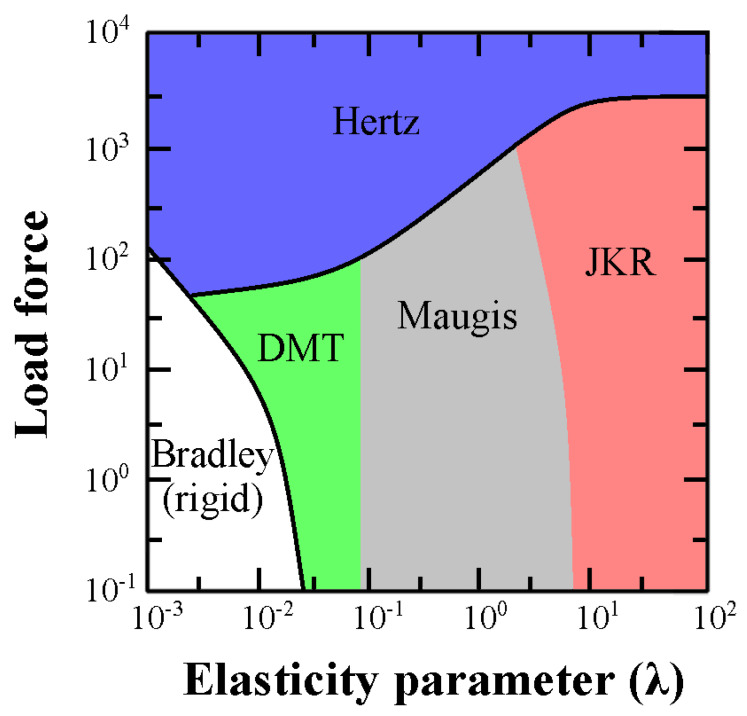
Map of the elastic behavior of matter according on the load force and λ parameter. In case of negligible adhesion, deformation falls in the Hertz limit. The DMT and JKR models are suitable for those samples that experiment small deformations or high adhesions, respectively. The Maugis theory is valid for the boundary region between the DMT and JKR models.

**Figure 3 nanomaterials-13-00963-f003:**
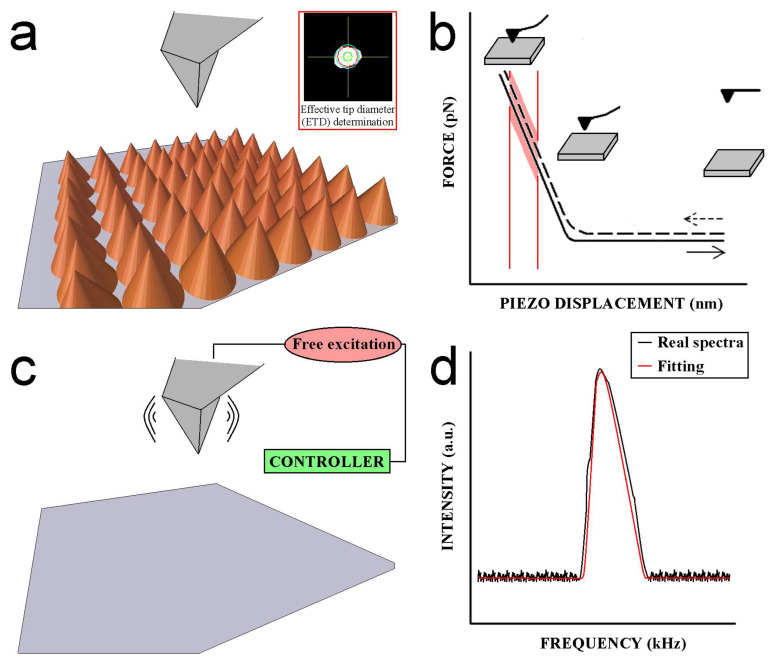
Schematic representation of (**a**) AFM tip radius quantification. The inset depicts the convolution performed to get the geometry and dimensions of the tip apex. (**b**) Typical force-curve acquired on stiff solid surfaces. The region comprised by the two red lines indicates the slope taken into account to calculate the AFM cantilever deflection sensitivity. (**c**) Brownian movement of the AFM cantilever when it is free of excitation. (**d**) Representative spectrum obtained by frequency sweep. The black line represents the raw experimental data, and the red line represents the fitting curve using simple harmonic oscillator models.

**Figure 4 nanomaterials-13-00963-f004:**
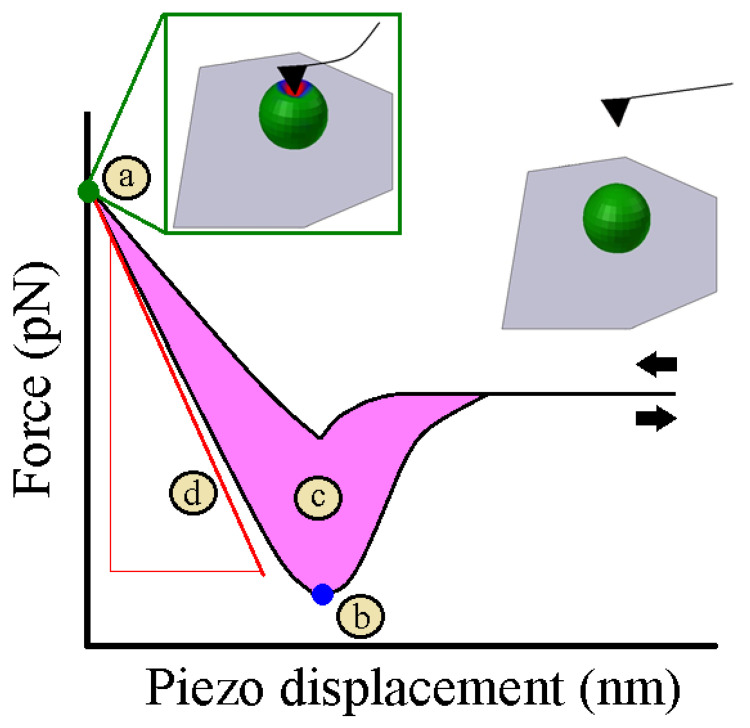
Mechanical properties extracted from the force–distance curve. (**a**) Tip sample contact point depicted by the green circle. (**b**) Tip sample unbinding point indicated by the blue circle. (**c**) Hysteresis between the approach and retract cycles highlighted by the pink shadow area. (**d**) Slope of the force–distance curve in the contact region shown by the red line. Black arrows indicate the direction of the AFM tip with respect to the sample surface.

**Figure 5 nanomaterials-13-00963-f005:**
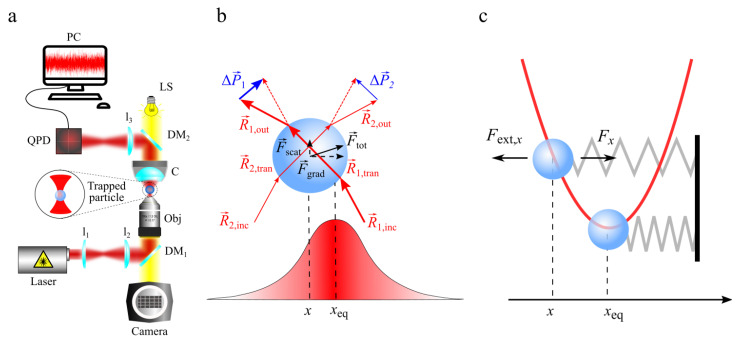
(**a**) Schematic representation of an OT setup. A laser beam, produced by a laser source is expanded by two lenses (l_1_ and l_2_) and reflected by a dichroic mirror (DM_1_) in order to overfill the back aperture of a high numerical aperture objective (Obj) producing a strong optical trap. A lens condenser (C) collects the interferogram arising from the interference between the light scattered by the particle and transmitted without interacting with the particle. The interferogram, containing the particle position information, is reflected by a second dichroic mirror (DM_2_) and projected by a lens (l_3_) to a QPD, which produces electrical signals proportional to the displacements of the particle within the optical trap. Analyzing these signals with a PC, we can rebuild the Brownian motion of the particle within the optical trap and calculate the trap stiffness. The dichroic mirrors DM_1_ and DM_2_ reflect the laser light, allowing the illumination of the particle by a light source (LS) and the observation of the particle dynamic by a camera. (**b**) GO approximation. Continuous red arrows represent light rays carrying a portion of the total optical power and linear momentum. Two different focused rays, ${\vec{R}}_{1,\mathrm{inc}}$ and ${\vec{R}}_{2,\mathrm{inc}}$, impinge on a spherical particle displaced by its equilibrium position. The ray ${\vec{R}}_{1,\mathrm{inc}}$ is thicker than ${\vec{R}}_{2,\mathrm{inc}}$ because it carries a higher amount of power and momentum, ${\vec{R}}_{1,\mathrm{inc}}>{\vec{R}}_{2,\mathrm{inc}}$, according to the intensity gradient of the focused laser beam. Once ${\vec{R}}_{1,\mathrm{inc}}$ and ${\vec{R}}_{2,\mathrm{inc}}$ cross the particle surface, they are transmitted through the particle as ${\vec{R}}_{1,\mathrm{tran}}$ and ${\vec{R}}_{2,\mathrm{tran}}$ and then transmitted out as ${\vec{R}}_{1,\mathrm{out}}$ and ${\vec{R}}_{2,\mathrm{out}}$. Blue arrows represent the momenta exchanged $\Delta {\vec{P}}_{1}>\Delta {\vec{P}}_{2}$ between the rays and the particle, calculated as the differences between the momenta associated to ${\vec{R}}_{i,\mathrm{inc}}$ and ${\vec{R}}_{i,\mathrm{out}}$, i.e., $\Delta {\vec{P}}_{i}={\vec{P}}_{i,\mathrm{inc}}-{\vec{P}}_{i,\mathrm{out}}$, with i=1,2. Dashed red arrows represent the shifted rays ${\vec{R}}_{1,\mathrm{inc}}$ and ${\vec{R}}_{2,\mathrm{inc}}$ used to calculate $\Delta {\vec{P}}_{1}$ and $\Delta {\vec{P}}_{2}$. The black continuous arrow represents the total optical force produced by $\left(\Delta {\vec{P}}_{1}+\Delta {\vec{P}}_{2}\right)$ during a time interval Δt, i.e., ${\vec{F}}_{\mathrm{tot}}=\left(\Delta {\vec{P}}_{1}+\Delta {\vec{P}}_{2}\right)$ /Δt. Black dashed arrows represent the component of ${\vec{F}}_{\mathrm{tot}}$ along the direction transversal to the laser beam propagation ${\vec{F}}_{\mathrm{grad}}$ and the component along the propagation of the laser beam, respectively, ${\vec{F}}_{\mathrm{scat}}$. (**c**) Schematic representation of an external force, $F_{\mathrm{ext},x}$, acting on a spherical particle and a restoring optical force $F_{x}$ (black arrows). Whenever $F_{\mathrm{ext},x}$ displaces a trapped particle from $x_{\mathrm{eq}}$ to a generic position x, an optical restoring force $F_{x}$, having the same magnitudo of $F_{\mathrm{ext},x}$ but opposite direction, takes place acting like a Hookean spring. Light gray lines represent Hookean springs connected to a solid origin (vertical black line). The red line represents the harmonic optical potential.

**Figure 6 nanomaterials-13-00963-f006:**
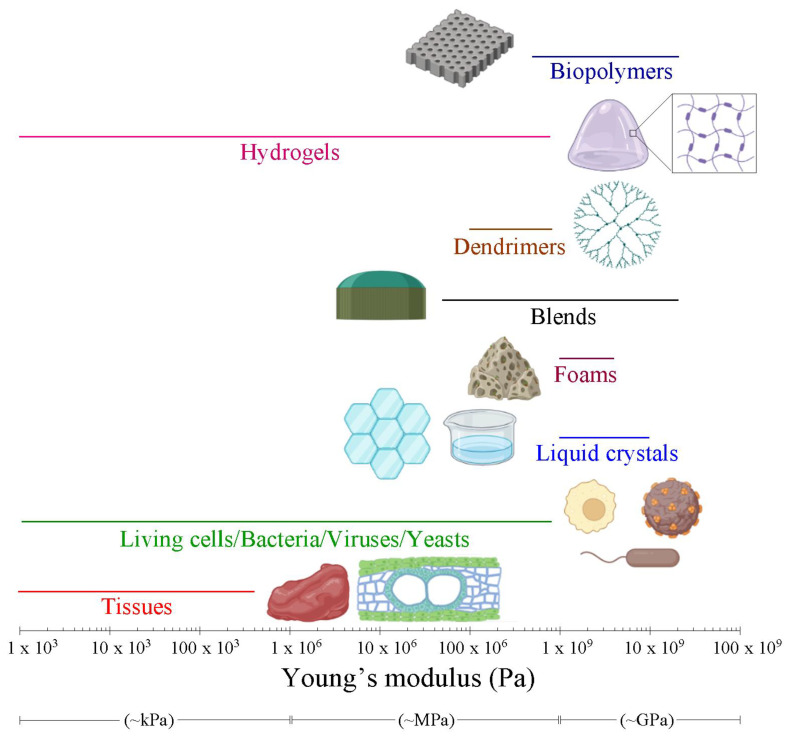
Elastic modulus ranges covered by the soft matter systems expressed in semilogarithmic scale. Images were created using BioRender.com.

**Figure 7 nanomaterials-13-00963-f007:**
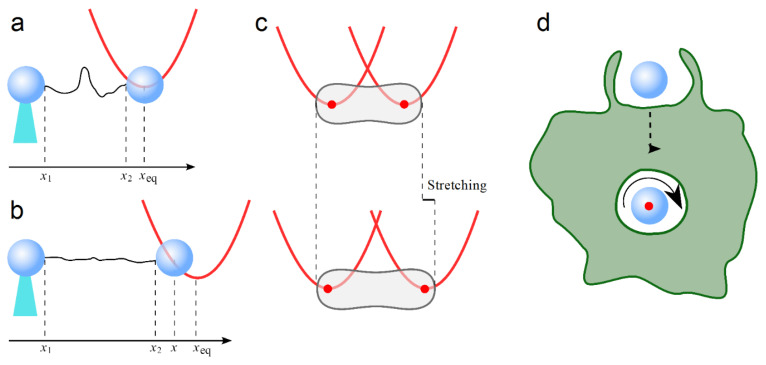
Stretching of a single DNA molecule with OT. The two ends of the molecule (black line) are attached to two colloidal spherical particles; one is fixed by a micropipette tips (cyan truncated triangle) and the other one is confined within an optical potential (red curve). (**a**) When the extension of the DNA molecule is shorter than $l_{0}$, i.e., $\left(x_{2}-x_{1}\right)<l_{0}$, a very small force is need to extend the polymer chain by pulling the trapped particle and eliminating the bends; the trapped particle remains almost centered at its equilibrium position, $x_{\mathrm{eq}}$. Where $x_{1}$ and $x_{2}$ are the positions of the two ends of the DNA molecule, $x_{1}$ is fixed, while $x_{2}$ changes according to the position of the trapped particle. (**b**) When $\left(x_{2}-x_{1}\right)\to l_{0}$, the DNA molecule shows an elastic stiffness, $k_{0}$, and the stretching force increases with the stretching. In this case, the trapped bead is displaced by $\Delta x_{\mathrm{eq}}=\left(x-x_{\mathrm{eq}}\right)$ from its equilibrium position. The position $x_{1}\;$ is experimentally set and fixed at the beginning of an experiment; $x_{2}$ can be easily calculated from the particle position x knowing the size of the bead; *x*_eq_ and the stretching force can be obtained by OT calibration. (**c**) Direct tapping of an RBC without any attached bead before (top) and during stretching (bottom). A two-beam OT is used to trap the two extremities of an RBC; one trap is fixed, while the other one is moved to stretch the cell; red spots indicate the position of focused laser beams. (**d**) Sketch of a macrophage internalization of a vaterite microspheres; the red spot indicates the laser beam, and the curved arrow indicates the direction of the particle rotation.

**Table 1 nanomaterials-13-00963-t001:** Hollomon, Swift and Voce equations to assess the ultimate tensile strength and the yield strength parameters.

Equation	Ultimate Tensile Strength	Yield Strength
Hollomon	$\sigma_{u}=K_{\;}n_{H}^{n_{H}}$	$\sigma_{s}=K_{H\;}{\left(0.002\right)}^{n_{H}}$
Swift	$\sigma_{u}=K{\left(n_{S}-\varepsilon_{0}\right)}^{n_{S}}$	$\sigma_{s}=K_{s\;}\varepsilon_{0}^{n}$
Voce	$\sigma_{u}=\frac{\sigma_{0}\;\beta }{\left(1+\beta \right)}$	$\sigma_{s}=\sigma_{0}-\sigma_{0}A$

where *σ_u_* is the ultimate tensile strength, *σ_s_* is the yield strength, *σ*_0_ is the saturation stress, *K* is strength coefficient, *n* is strength-hardening parameter, *ε*_0_ is a potential pre-strain parameter, *A* and *β* are material coefficients. Subscripts refer to the equation they are based upon. Then, Young’s modulus can be obtained from Equation (1). Tensile testing experiments have been carried out to address the mechanical properties of biodegradable cornstarch- [112], starch/dolomite- [113] or lignocellulosic- [114,115] based polymers, hydrogels made by carbon dots, hydroxyapatite and polyvinyl acetate [116] or chitosan-poly (acrylic acid-co-acrylamide) double network [117], natural-rubber-modified flame-retardant organic montmorillonite [118] or chlorhexidine-loaded poly (amido amine) [119] dendrimers, blends consisting of fibrillar polypropylene and polyethylene terephthalate [120] or poly ε-caprolactone/poly-(lactide-co-ε-caprolactone (PLCL) [121], polyurethane [122] and polyethylene [123] foams, organosilicone elastomer liquid crystals [124], and skeletal muscle tissues [125] or PLCL layered sheets with mesenchymal stem cells [126].

**Table 2 nanomaterials-13-00963-t002:** The most convenient AFM cantilever spring constants related to the expected Young’s modulus of soft matter samples.

Spring Constant (k)	Young’s Modulus
0.5 N/m	1 MPa–20 MPa
5.0 N/m	5 MPa–500 MPa
40.0 N/m	200 MPa–2 GPa
200.0 N/m	1 GPa–20 GPa
350.0 N/m	10 GPa–100 GPa

**Table 3 nanomaterials-13-00963-t003:** Elastic modulus of illustrative soft matter samples measured recently by AFM. Samples are sorted by alphabetical order and classified inside the soft matter system they belong. (*) Bis = bis-acrylamide. Room conditions refer those AFM measurements conducted in air at environmental temperature and relative humidity (R.H.).

Soft Matter System	Sample	Conditions	Elastic Modulus	[Ref.]
Biopolymer	Cellulose nanocrystals (CNCs) films	45% R.H. (L.F) 200 nN	10.3 ± 0.9 GPa	[168]
Biopolymer	Lignin films	45% R.H (L.F) 200 nN	6.3 ± 0.4 GPa	[168]
Biopolymer	Oxidized lignin films	45% R.H (L.F) 200 nN	11.0 ± 1.6 GPa	[168]
Polymer	Polypropylene (PP)	50 °C	1.4 ± 0.1 GPa	[43]
Polymer	Polybutylene succinate (PBS)	50 °C	548.3 ± 14.2 MPa	[43]
Hydrogel	Acrylamide (5.5%)–Bis(*) 0.03%	Room	2.0 ± 0.1 kPa	[169]
Hydrogel	Acrylamide (12.0%)–Bis(*) 0.15%	Room	29.0 ± 6.2 kPa	[169]
Hydrogel	Chitosan—genipin	pH 3, 1 h react. t.	477 MPa	[170]
Hydrogel	Chitosan—genipin	pH 6, 1 h react. t.	615 MPa	[170]
Dendrimer	5 poly(amido amine) (PAMAM)	Room	700 MPa	[171]
Dendrimer	Polyelectrolite microcapsules	Room	150 MPa	[172]
Blend	CNC:oxidized lignin	45% R.H (L.F) 200 nN	13.6 ± 0.6 GPa	[114]
Blend	Polyimide:graphene oxide	Load force (L.F) 55 µN	6.3 ± 0.5 GPa	[173]
Blend	Polyurethane:carbon	100 impulses, 0.5 keV	75 MPa	[174]
Foam	Polyisocyanurate	Room. Height 30 mm	3.4 ± 0.4 GPa	[175]
Liquid crystal	Poly (p-phenylene terephthalamide)	Load force 1000 µN	5.6 GPa	[176]
Biological (bacteria)	*Staphylococcus epidermidis*	Deionized water	1.0 ± 0.3 MPa	[162]
Biological (bacteria)	*Staphylococcus epidermidis*	100 mM CaCl_2_	0.6 ± 0.2 MPa	[162]
Biological (living cell)	Human osteosarcoma	Liquid	34.3 ± 2.4 kPa	[177]
Biological (living cell)	Human skin (normal)	Room L.F 2.9 µN	401 ± 148 MPa	[166]
Biological (living cell)	Patient skin (benign nevus)	Room L.F 2.9 µN	575 ± 107 MPa	[166]
Biological (living cell)	Patient skin (melanoma)	Room L.F 2.9 µN	188–787 MPa	[166]
Biological (plant)	Pollen tube *Arabidopsis thaliana*	Room L.F 6.5 nN	46 ± 12 MPa	[178]
Biological (tissue)	Human colon cancer	Fixed-frozen section	115.8 kPa	[164]
Biological (tissue)	Mice cortical bone	PBS. L.F 0.5 nN	0.86 kPa	[167]
Biological (virus)	HK97 bacteriophage	Ident. depth 5.5 nm	400 MPa	[165]
Biological (virus)	HK97 bacteriophage	Ident. depth 8.5 nm	900 MPa	[165]
Biological (virus)	Zika viral particles	Room L.F 6.0 nN	234 kPa	[179]
Biological (yeast)	*Saccharomyces cerevisiae*	Room. L.F 1.0 µN	5.1 ± 1.5 MPa	[180]

**Table 4 nanomaterials-13-00963-t004:** DNA elasticity parameters for different multivalent cation buffer solutions. Reprinted/adapted with permission from Ref. [181]. Copyright 1997, Elsevier.

Buffer Composition	$l_{p}$ (nm)	$k_{0}$ (pN)
10 mM Na^+^ (NaHPO_4_ buffer, pH 7.0)	47.4 ± 1.0	1008 ± 38
150 mM Na^+^, 5 mM Mg^2+^ (NaHPO_4_ buffer, pH 7.0)	43.1 + 1.3	1205 ± 87
10 mM Na^+^, 100, LM spermidine (NaHPO_4_ buffer, pH 7.0)	38.7 ± 1.0	1202 ± 83
20 mM Tris, 130 mM K^+^, 4 mM Mg^2+^ (PTC buffer, pH 8.0)	41.0 ± 0.8	1277 + 57

**Table 5 nanomaterials-13-00963-t005:** DNA elasticity parameters for different Na^+^ cation concentrations. Reprinted/adapted with permission from Ref. [182]. Copyright 2002, Elsevier.

Na^+^ Concentration (mM)	$l_{p}$ (nm)	$k_{0}$ (pN)
2.6	68 ± 2	741 ± 56
10	67 ± 4	741 ± 147
25	58 ± 3	790 ± 104
53.5	52 ± 1	1078 ± 64
100	48 ± 2	884 + 116
250	46 ± 1	1038 ± 69
500	47 + 2	1049 ± 226
1000	46 ± 2	1256 ± 217

**Table 6 nanomaterials-13-00963-t006:** Elasticity parameters of λ-bacteriophage DNA for different ionic strengths of NaCl concentration. Reprinted/adapted with permission from Ref. [186]. Copyright 1997, Natural Academy of Sciences.

Ionic Stength (mM)	$l_{p}$ (nm)	$k_{0}$ (pN)
1.86	94.9 ± 5.9	649 ± 82
3.72	75.7 ± 2.5	745 ± 100
5.58	76.7 ± 5.4	476 ± 142
7.44	62.2 ± 3.7	686 ± 65
93.0	65.2 ± 2.7	452 ± 35
18.6	52.9 ± 9.5	532 ± 67
93.0	51.1 ± 1.8	1006 ± 2
186	52.5 ± 12.4	1401 ± 313
586	55.9 ± 3.2	1435 ± 160

**Table 7 nanomaterials-13-00963-t007:** Stretching parameters and deformability index of RBCs obtained by dual-beam optical tweezers for healthy control subjects and study subjects affected by diabetes mellitus (DM) and diabetic retinopathy (DR). Data are obtained with 925 cell cycles (5–10 cycles for single cell) [189].

	Control Group	DM Group	DR Group
Average unstretched cell size (μm)	8.45 ± 0.25	8.68 ± 0.49	8.82 ± 0.32
Average maximal stretched cell size (μm)	9.04 ± 0.17	9.23 ± 0.49	9.39 ± 0.26
Average difference between stretched and unstretched cell size (μm)	0.59 ± 0.19	0.56 ± 0.32	0.56 ± 0.24
Deformability index	0.0698 ± 0.024	0.0645 ± 0.03	0.0635 ± 0.029

## Data Availability

Additional data are available upon reasonable request.

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
