# Peer review of "Investigation of Soft Matter Nanomechanics by Atomic Force Microscopy and Optical Tweezers: A Comprehensive Review"

_nanomaterials, 2023, doi:10.3390/nano13060963_

Round 1
Reviewer 1 Report
See the attached file.

Author Response
The authors would like to sincerely acknowledge to the Reviewer for the time spent in the thoughtful assessment of the present manuscript entitled “Comprehensive review – Potential of atomic force microscopy and optical tweezers to address the nanomechanical properties of the soft matter” (Manuscript ID nanomaterials-2212305). All suggestions have been carefully considered and subsequently covered during the manuscript main body text.
The title was shortened as “Investigation of soft matter systems by atomic force microscopy and optical tweezers: A comprehensive review”. We consider the current title will be easier to remember by the readers and straightforward to appear in scientific databases. Moreover, the entire manuscript body text was thoroughly checked by us.
Please find below the responses to the Reviewers queries attached point-by-point. The statements included in the updated review version are highlighted in red colour.
The manuscript number nanomaterials-2212305, entitled “Comprehensive review – Potential of atomic force microscopy and optical tweezers to ad dress the nanomechanical properties of the soft matter” presents atomic force microscopy (AFM) and optical tweezers (OT) high-resolution techniques as nanotechnology tools to achieve the biomechanical parameters at the single molecule level, compared to other classical bulk methods. Also recent examples of nanomechanical measurements performed by AFM and OT in hydrogels, biopolymers and cellular systems are provided in the review manuscript. The idea of the study is interesting. The title was carefully chosen, the topic of the review being suitable for Nanomaterials journal.
Response: We sincerely acknowledge the positive comments provided by Reviewer1 regarding our scientific manuscript.
The Abstract is well written, but the English must be revised. For example for: Soft matter exhibits multitude of intrinsic physico-chemical attributes being their mechanical properties one of determinant characteristics to define their performance. the alternative can be: Soft matter exhibits a multitude of intrinsic physico-chemical attributes, their mechanical properties being one of determinant characteristics to define their performance.
Response: The Abstract section was modified according to the suggestion of Reviewer1.
In the Introduction part the framework and hypotheses were clear, emphasizing the types of soft matter systems, the techniques to determine the mechanical properties of soft matter and the single molecule techniques like atomic force microscopy (AFM) and optical tweezers (OT).
Response: We are pleased that Reviewer1 enjoyed our work and the structure followed in the manuscript.
Next, the review is divided in the following sections: II) Mechanical properties (explain the mechanical properties – Young’s modulus (E), hardness (H), viscoelasticity, fracture toughness and viscous energy dissipation – susceptible to be measured in soft matter systems) Observations: In eqn 3 (I suggest to name the equations with eq., not with eqn.), are explained some factors like Ei.relax the elastic moduli related for the relaxation event and F(t) the time-dependent relaxation function, but there are not part of the equation 3.
Response: We agree with the suggestion pointed out by Reviewer1. For this reason, the abbreviation terms for all the equations were modified accordingly. Furthermore, we introduced the equation related to the relaxation phenomena at lines 210-212 of the text.
“The main linear viscoelasticity model was defined by Volterra equations [87] (eqn., 3 and eq., 4, respectively):”
(eq., 4)”.
Furthermore, it was also detailed the relation between both terms: “The Ei.creep and Ei.relax are related through their respective time dependent functions (K(t) and F(t), respectively):
(eq., 5)
Finally, the compliance functions of the Eicreep and Eirelax, D(t) and R(t) are correlated by the following expression:
(eq., 6)”.
III) Non-nanotechnology techniques to determine mechanical properties, - Multifrequency magnetic resonance elastography (MRE) Observations: You mention the loss angle (φ), but is not present in eqn.8 and eqn. 9.
Response: We appreciate this comment from Reviewer1. The text indicates that during the multifrequency magnetic resonance elastography measurements the shear modulus ( and loss angle (φ) parameters are simultaneously obtained. The loss angle does not contribute in the direct estimation of the elastic modulus, but it is related to the calculation of the loss modulus (G´´). Where G´´ represents the viscous response of the measuring energy dissipated per cycle of sinusoidal deformation. G and G´´ are related by the expression below described:
We agree with Reviewer1 that this information could be interesting for the potential readers because it directly correlates the interplay of both modulus. Furthermore, G´´ exemplifies the viscous performance of the tested soft matter. For all these reasons, we added the equation and the following statement at lines 295-297: “The loss modulus (G´´) can be calculated through the φ and the previously estimated G:
The viscous response of soft matter samples is determined by the extension of G´´.”
Observations: Please rename subsections 6.1. and 6.2., which are the same as 5.1. and 5.2. You can use 6.1. Atomic force microscopy to evaluate elastic properties... and 6.2. Elasticity of soft matter system evaluated by Optical tweezers ....for example.
Response: We thank Reviewer1 for his suggestion. The subsections 6.1 and 6.2 were renamed according this suggestion from Reviewer 1: “6.1. Atomic force microscopy (AFM) to evaluate elastic properties of the soft matter” and “Elasticity of soft matter systems addressed by optical tweezers (OT)”, respectively.
The figures and tables are inspired and help the reader to understand better the phenomena. The manuscript is well written and documented with updated bibliographic references, among them few of the authors. The chosen references are relevant for the approached subject.
Response: We once again thank Reviewer1 for the assessment of our manuscript and for all his comments which have significantly contribute to improve the scientific content quality of the manuscript.
We expect the present revised review version and the rebuttal answers settle all the comments, doubts and suggestions given by the reviewers.
The submitting authors accept responsibility for the following:
- We have the consent from all authors to submit the manuscript and all authors accept complete responsibility for the contents of the manuscript.
- The manuscript is not currently under consideration elsewhere and the work reported will not be submitted for publication elsewhere until a final decision has been made as to its acceptability by the Journal.
- The manuscript is truthful original revision.
Yours sincerely,
ALESSANDRO MAGAZZÙ and CARLOS MARCUELLO

Reviewer 2 Report
The authors present a review of two methods for investigating soft matter mechanical properties at the nanoscale. Atomic force microscopy and optical tweezers are reviewed. The authors should consider the following matters when preparing a revision.
1. In the Introduction, the information associated with each underlined word could instead be broken out into a bulleted list. Doing so would make the manuscript easier to read by breaking up the very long first paragraph. Alternatively, making each underlined item a new paragraph could be a way to improve the readability of the first paragraph.
2. There are places where the authors attempt more complex language usage than is probably necessary. Notably, there are instances, such as Introduction lines 119-127, where trying to employ complex phrasing results in sentences that are very badly formed and confusing. Employing a more succinct style would improve the readability of the review.
3. There are multiple instances of the terms in the equations in Section 2 not being suitably described.
4. Sections 2, 3 and 4 are not entirely necessary in light of the intended topic of the review. The authors should focus on the nanoscale properties and the two methods that they are reviewing. The material on the macroscopic quantities and how they are measured is an unnecessary distraction unless the authors do a better job of connecting the parameters normally applied to bulk materials to what they mean at the nanoscale. It is relatively easy to understand for the Young’s modulus, but the other properties are not as readily related and do not seem to come up later in the document.
5. Section 5.1 is useful information, but it seems out of place before a description of what mechanical properties can be measured using AFM and optical tweezers and a review of examples of their use. Section 5.2 does not feel out of place, but it suffers from not describing what can be measured with optical tweezers.
6. Section 5.1 is difficult to follow due to the ordering of some of the sentences. The first paragraph is also very long.
7. The sections on AFM contain a large number of technical “how to” details that are not in the optical tweezers sections. These sections are considerably harder to read, so it is not clear that the presence of this information is beneficial.
8. A large fraction of the second paragraph of Section 7 seems better suited to a section prior to the final one.
8. Even without reading the author list, the reader is left with a very clear impression that there were at least two authors of the paper due to considerable differences in writing style and quality in different sections. Effort should be made to reach a consistent style.
Author Response
We thank the reviewer for recognizing the importance of the article entitled “Comprehensive review – Potential of atomic force microscopy and optical tweezers to address the nanomechanical properties of the soft matter” (Manuscript ID nanomaterials-2212305) and for making excellent suggestions.
The title was shortened as “Investigation of soft matter systems by atomic force microscopy and optical tweezers: A comprehensive review”. We consider the current title will be easier to remember by the readers and straightforward to appear in scientific databases. Moreover, the entire manuscript body text was thoroughly checked by us.
We revised the article as per the reviewer's suggestions. Please see below our response to the comments point-by-point. The statements included in the updated review version are highlighted in red colour.
The authors present a review of two methods for investigating soft matter mechanical properties at the nanoscale. Atomic force microscopy and optical tweezers are reviewed. The authors should consider the following matters when preparing a revision.
- In the Introduction, the information associated with each underlined word could instead be broken out into a bulleted list. Doing so would make the manuscript easier to read by breaking up the very long first paragraph. Alternatively, making each underlined item a new paragraph could be a way to improve the readability of the first paragraph.
Response: We sincerely acknowledge Reviewer2 for this comment. We broke the introduction out into a bullet list as suggested
- There are places where the authors attempt more complex language usage than is probably necessary. Notably, there are instances, such as Introduction lines 119-127, where trying to employ complex phrasing results in sentences that are very badly formed and confusing. Employing a more succinct style would improve the readability of the review.
We agree with the comment of Reviewer2 and we rephrased the Introduction lines 137-151 as follows “AFM shows many advantages in comparison to other techniques such as:
- the possibility of investigating samples in liquid environments, mimicking the inner cellular conditions allowing in vivo In doing this, by AFM we can measure the mechanical properties of the soft matter as function of the pH or ionic strength of the liquid media [70];
- AFM measurements can be performed under suitable conditions to preserve the integrity of the investigated sample, conversely to cryo-transmission electron microscopy (cryo-TEM) where ultra-low temperatures are required [71].
AFM does not require to stain the sample with contrast agents conversely to other techniques like scanning electron microscopy (SEM) or TEM, avoiding any artefacts and interferences with the investigated properties [72]. ”.
- There are multiple instances of the terms in the equations in Section 2 not being suitably described.
Response: We thank Reviewer2 for pointing this issue out. The missing terms have been carefully revised and added as described in the response to Reviewer1.
- Sections 2, 3 and 4 are not entirely necessary in light of the intended topic of the review. The authors should focus on the nanoscale properties and the two methods that they are reviewing. The material on the macroscopic quantities and how they are measured is an unnecessary distraction unless the authors do a better job of connecting the parameters normally applied to bulk materials to what they mean at the nanoscale. It is relatively easy to understand for the Young’s modulus, but the other properties are not as readily related and do not seem to come up later in the document.
Response: We thank Reviewer2 for the keen observation. This review is mainly focused on the determination of mechanical properties by nanotechnology tools like AFM and OT. Nevertheless, we consider opportune to introduce the physical basis of these mechanical properties (section 2), to discuss about other about techniques to measure the bulk mechanical properties (section 3) and about the phenomenological models to assess the sample Young’s modulus (section 4). It is of a paramount importance to provide a physical background and overview of all macroscopic mechanical properties before introducing or to better understanding the nanoscopic mechanical properties. Then, section 4 is directly related to the AFM field because the models here discussed are required to transform the force exerted by the AFM tip apex on the sample surface depending on the AFM tip geometry.
Section 2 already ends with this sentence that directly links with the content showed in the next sections: “The study of mechanical properties of soft matter systems is fundamental to better understand their nature which could assist to find future potential social and industrial applications”. Additionally the following statement was added (lines 270-275): “Next sections will provide the fundamental knowledge to the readers about the current available techniques to determine the mechanical properties at bulk and nanoscale levels. Then, the review will focus on the Young’s modulus assessment by nanotechonology tools like AFM and OT. For it, the existing physical models to extract the elastic modulus of the tested soft matter samples will be discussed depending on the AFM tip geometry and the forces involved during the conducted force-distance curves.”.
Then, the following statements were added to highlight the importance of these sections and to establish a better connection between them as suggested by Reviewer2:
“The main limitation of the above described non-nanotechnology tools is their inability to observe singular events or mechanical gradients in bulk measurements. Furthermore, the continuous necessity of measuring the elastic properties of specific local areas drove to the progress of single molecule techniques in-fine contribution to the physical models described in the next section” (at the end of the section 3, lines 373-377).
“This section explains all the existing theoretical frameworks to extract the elastic modulus of soft matter samples by nanotechnology tools, overall by AFM where the tip apex works as a nanoindenter” (at the beginning of the section 4, lines 384-385).
- Section 5.1 is useful information, but it seems out of place before a description of what mechanical properties can be measured using AFM and optical tweezers and a review of examples of their use. Section 5.2 does not feel out of place, but it suffers from not describing what can be measured with optical tweezers.
Response: We appreciate this comment from Reviewer 2.
Section 5.1. (AFM) discuss all the aspects involved in the proper characterization of the AFM tip before the nanomechanical experiments. We believe that this aspect is fundamental since the accuracy of the obtained results directly relies on it. There are many cases in literature where the authors did not follow these steps (for instance, by not quantifying the AFM tip radius and in turn, using the nominal value provided by the supplier. This fact leads significant estimation errors). of the scope of the beginning of this section is to provide to the potential readers and newcomers in the field the working principles of the AFM and the strong capabilities to decipher the sample nanomechanical properties.
We agree with Reviewer2 with the need that further details must be provided in order to highlight the mechanical properties that can be assessed by AFM. For this reason, the following content was added at the end of the section 5.1 (lines 545-572):
“AFM is capable to record force-distance curves (Figure 4) at local sample areas discriminating between different mechanical properties.
Figure 4. Mechanical properties extracted from the force-distance curve. (a) Tip-sample contact point depicted by the green circle. (b) Tip-sample unbinding point indicated by the blue circle. (c) Hysteresis between the approach and retract cycles highlighted by the pink shadow area. (d) Slope of the force-distance curve in the contact region showed by the red line. Black arrows indicate the direction of the AFM tip respect to the sample surface.
During a single force-distance curve cycle the AFM tip approaches to the external sample surface until the contact between both bodies occurs (Fig. 4a). At this point, the rigid AFM tip apex causes a deformation of the indented surface. Then, a different direction of movement takes place and the tip moves away the sample. Adhesion forces between the tip and the sample emerged during the contact being broken at sudden retract point (Fig. 4b). The tip will move up to the original position defined at the start of the cycle. The energy dissipation and sample stiffness can be estimated once the force-distance curve is recorded. The area between the approach and retract curves determines the energy dissipation (Fig. 4c), whereas the stiffness relies on the slope of the force-distance curve (Fig. 4d). The stiffness of the soft matter can be quantitatively converted to the Young’s modulus through the mechanic models described in Section 4 which are selected depending on the geometry of the AFM tip and the nature of the forces that dominate during the contact. For all the aforementioned reasons, the accurate characterization of the used AFM tip is crucial to precisely determine the Young’s modulus of soft matter systems.
Finally, an array of force-distance curves over specific sample surface areas can lead three-dimensional maps where it is possible to simultaneously compare the topography of the scanned area with the elasticity map. This aspect allows to show local variations in surface Young’s modulus”.
Regarding the section 5.2 we added the following text at lines 661-666 in order to clarify what can be measured by OT and to link this section with section 6.2 where all the quantities and properties of soft matter system measured by OT are explained in details.
By OT we can measure an external force acting on a trapped bead and the resulting displacement from its equilibrium position. Moreover, OT can exert optical forces on soft matter systems and are currently employed to study the elastic properties of these systems by measuring the force required for their stretching. See section 6.2 for more details.
- Section 5.1 is difficult to follow due to the ordering of some of the sentences. The first paragraph is also very long.
Response: We agree with the Reviewer 2 suggestion. The first paragraph of section 5.1 was divided in four different parts:
- “Atomic force microscopy has shown (…) AFM tip sharpness”.
- “The effective tip diameter (ETD) (…) ( 3b)”.
- “Nevertheless, determination of (…) spring constants than 1 N/m”.
- “Finally, the thermal noise (…)”.
- The sections on AFM contain a large number of technical “how to” details that are not in the optical tweezers sections. These sections are considerably harder to read, so it is not clear that the presence of this information is beneficial.
Response: We appreciate this suggestion of Reviewer 2. As aforementioned stated, AFM is a tricky technique and we consider that these details will be beneficial for the potential readers and stakeholders. The English was subsequently revised in order to increase the manuscript readability.
- A large fraction of the second paragraph of Section 7 seems better suited to a section prior to the final one.
Response: First paragraph of Section 7 is focused on the discussion of this review work. Then, second paragraph underlines the promising open avenues of AFM and OT (specially combined with other techniques) to unravel not only the mechanical properties of the soft matter, but also other properties at the same time.
The second paragraph was split in the following three parts:
“Promising avenues of research are opened (…) of bone implants [211]” (specific for AFM).
“On the other hand, OT (…) exploited for sample imaging” (specific for OT).
“The combination of AFM and OT (…) potential samples of interest” (common for both).
- Even without reading the author list, the reader is left with a very clear impression that there were at least two authors of the paper due to considerable differences in writing style and quality in different sections. Effort should be made to reach a consistent style.
Response: We thank Reviewer2 for all the insightful comments which have greatly improve the manuscript quality. Additionally, the entire manuscript content was extensively checked by us.
We expect the present revised review version and the rebuttal answers settle all the comments, doubts and suggestions given by the reviewers.
The submitting authors accept responsibility for the following:
- We have the consent from all authors to submit the manuscript and all authors accept complete responsibility for the contents of the manuscript.
- The manuscript is not currently under consideration elsewhere and the work reported will not be submitted for publication elsewhere until a final decision has been made as to its acceptability by the Journal.
- The manuscript is truthful original revision.
Yours sincerely,
ALESSANDRO MAGAZZÙ and CARLOS MARCUELLO

Round 2
Reviewer 2 Report
Thank you for making changes to the manuscript in response to the recommendations. The authors are encouraged to perform additional proof-reading for language usage.
Author Response
We thank the Reviewer2 for recognizing the importance of the manuscript paper and for making excellent suggestions. We revised the article according the suggestion made by the Reviewer2. Please, see attached the manuscript revised version.
